# KLK6/PAR1 Axis Promotes Tumor Growth and Metastasis by Regulating Cross-Talk between Tumor Cells and Macrophages

**DOI:** 10.3390/cells11244101

**Published:** 2022-12-16

**Authors:** Yo Sep Hwang, Hee Jun Cho, Eun Sun Park, Jeewon Lim, Hyang Ran Yoon, Jong-Tae Kim, Suk Ran Yoon, Haiyoung Jung, Yong-Kyung Choe, Yong-Hoon Kim, Chul-Ho Lee, Yong Tae Kwon, Bo Yeon Kim, Hee Gu Lee

**Affiliations:** 1Immunotherapy Research Center, Korea Research Institute of Bioscience and Biotechnology, Daejeon 34141, Republic of Korea; 2Department of Biomolecular Science, University of Science and Technology (UST), Daejeon 34113, Republic of Korea; 3Laboratory Animal Resource Center, Korea Research Institute of Bioscience and Biotechnology, Daejeon 34141, Republic of Korea; 4Protein Metabolism Medical Research Center, Department of Biomedical Science, College of Medicine, Seoul National University, Seoul 110-799, Republic of Korea; 5Anticancer Agent Research Center, Korea Research Institute of Bioscience and Biotechnology, Ochang, Cheong won, Cheongju 28116, Republic of Korea

**Keywords:** KLK6, PAR1, tumor microenvironment, macrophage, metastasis

## Abstract

Kallikrein-related peptidase (KLK)6 is associated with inflammatory diseases and neoplastic progression. KLK6 is aberrantly expressed in several solid tumors and regulates cancer development, metastatic progression, and drug resistance. However, the function of KLK6 in the tumor microenvironment remains unclear. This study aimed to determine the role of KLK6 in the tumor microenvironment. Here, we uncovered the mechanism underlying KLK6-mediated cross-talk between cancer cells and macrophages. Compared with wild-type mice, KLK6−/− mice showed less tumor growth and metastasis in the B16F10 melanoma and Lewis lung carcinoma (LLC) xenograft model. Mechanistically, KLK6 promoted the secretion of tumor necrosis factor-alpha (TNF-α) from macrophages via the activation of protease-activated receptor-1 (PAR1) in an autocrine manner. TNF-α secreted from macrophages induced the release of the C-X-C motif chemokine ligand 1 (CXCL1) from melanoma and lung carcinoma cells in a paracrine manner. The introduction of recombinant KLK6 protein in KLK6−/− mice rescued the production of TNF-α and CXCL1, tumor growth, and metastasis. Inhibition of PAR1 activity suppressed these malignant phenotypes rescued by rKLK6 in vitro and in vivo. Our findings suggest that KLK6 functions as an important molecular link between macrophages and cancer cells during malignant progression, thereby providing opportunities for therapeutic intervention.

## 1. Introduction

The tumor microenvironment is the region surrounding the tumor and encompasses immune cells, stromal cells, blood vessels, secreted factors, and the extracellular matrix [1]. The reciprocal interaction between cancer cells and nonmalignant surrounding cells significantly affects tumor progression by influencing proliferation, local invasion, and metastatic dissemination [2,3]. Immune cells in the tumor microenvironment such as tumor-associated macrophages (TAMs), dendritic cells, myeloid-derived suppressor cells, T cells, mast cells, and natural killer cells play critical roles in tumor progression [4,5]. Macrophages are the most abundant host immune system- associated stromal cells in multiple malignancies [6]. TAMs regulate tumor growth, angiogenesis, metastasis, and drug resistance by producing cytokines such as tumor necrosis factor-alpha (TNF-α), interleukin (IL)-6 and IL-1β, and chemokines such as CXC chemokine ligand 1 (CXCL1) and CXCL2 [7]. TNF-α is mostly produced by macrophages but also by other immune cells and non-immune cells such as endothelial cells and smooth muscle cells. TNF-α secreted by macrophages promotes the expression of CXCL1, which is involved in cancer development and malignant progression [8].

Kallikrein-related peptidase (KLK)6 belongs to a family of serine proteases with trypsin- or chymotrypsin-like activities [9]. KLK6 is expressed in many tissues and participates in various physiological and pathological processes. It can degrade components of the extracellular matrix such as fibrinogen and various collagens [10,11,12]. KLK6 also triggers intracellular signaling by the activation of protease-activated receptors (PARs) that are implicated in various biological processes including inflammatory responses and cancer progression [13,14,15]. Deregulation of KLK6 expression is observed in inflammatory and neurodegenerative diseases. High levels of the KLK6 protein have been detected in individuals with various skin diseases such as psoriasis and atopic dermatitis, and rheumatoid arthritis; KLK6 promotes psoriasis dermatitis and inflammatory joint disease via PAR1 signaling [16,17,18,19,20]. It also promotes the degeneration of cerebellar neurons and exacerbates glutamate neurotoxicity through PAR1 and PAR2 [21].

Several studies have suggested that the KLK6 transcript and protein are highly expressed in various human cancers including breast, lung, pancreatic, colorectal, gastric, and ovarian cancers [22,23,24,25,26,27]. Enhanced KLK6 expression in skin cancer is associated with malignant progression. Ectopic KLK6 expression in a mouse keratinocyte cell line was found to induce spindle-like morphology and promote cell proliferation, migration, and invasion [28]. On the other hand, KLK6 expression is not detected in melanomas but is observed in keratinocytes and stromal cells adjacent to benign nevi, primary melanomas, and cutaneous metastatic lesions [29]. A recent study showed that KLK6 deficiency mediates the resistance to skin tumor development and that this resistance may be associated with reduced 7,12-dimethylbenz[a]anthracene/12-O-tetradecanoylphorbol 13-acetate (DMBA/TPA)-induced inflammation [30]. These studies suggest paracrine effects of extracellular KLK6 during tumor development and malignant progression. However, the pathogenic role of KLK6 in the tumor microenvironment remains unclear; moreover, little is known about how KLK6 contributes to cross-talk between cancer cells and inflammatory cells.

This study aimed to determine the role of KLK6 in the tumor microenvironment. KLK6 promoted the production of TNF-α from macrophages via PAR1. TNF-α secreted by macrophages stimulated CXCL1 production in melanoma cells to facilitate tumor growth and metastasis. Our findings suggest that KLK6 functions as an important molecular link between macrophages and cancer cells during malignant progression.

## 2. Materials and Methods

### 2.1. Reagents and Antibodies

Anti-KLK6 rabbit polyclonal antibody was developed through Abclon (Guro-gu, Seoul, Republic of Korea). TNF-α (Cell Signaling Technology, CST, Danvers, MA, USA, CST-11948), Interleukin (IL)-1β (CST, 12426), TNF-α neutralizing antibody (CST, 11969), VEGF (CST, 50661), and cleaved caspase-3 (CST, 9661) antibodies were purchased from Cell Signaling Technology. CXCL1 neutralizing antibody (R&D Systems, Minneapolis, MN, USA, MAB453), CXCL1 (R&D Systems, AF-453), PAR1 antagonist (Med-ChemExpress, Monmouth Junction, NJ, USA, SCH 79797), and PAR2 antagonist (Axon-Medchem, Reston, VA, USA, Axon 1622) were purchased from R&D Systems, MedChemExpress and Axon-Medchem, respectively. β-actin (Santa Cruz Biotechnology, Santa Cruz, CA, USA, sc-47778), Ki67 (Santa Cruz, sc-550609), GFP (Santa Cruz, sc-9996), PAR1 (Novus Biologicals, Novus, Centennial, CO, USA, NBP1-71770), and PAR2 (Abcam, Cambridge, UK, ab180953) antibodies were obtained from Santa Cruz Biotechnology, Novus Biologicals and Abcam, respectively. β-actin was used to normalize the level of proteins. Recombinant murine TNF-α (PeproTech, Rocky Hill, NJ, USA, 315-01), IL-1β (PeproTech, 211-11), IFN-γ (PeproTech, 315-05), and CXCL1 (PeproTech, 250-11) were purchased from PeproTech.

### 2.2. Cell Culture

Mouse macrophage cell lines RAW 264.7 (TIB-71) and a melanoma cell line B16F10 (CRL-6475) and Lewis lung carcinoma (LLC, CRL-1642) were purchased from the American Type Culture Collection (ATCC, Rockville, MD, USA). All cell lines were cultured in Dulbecco’s modified Eagle’s medium (DMEM; Gibco, MA, USA) supplemented with 10% Fetal Bovine Serum (FBS; HyClone laboratories, Logan, UT, USA) and antibiotics (100 U/mL penicillin and 100 µg/mL streptomycin (Gibco-BRL). The cells were incubated in a humidified incubator with 5% CO_2_ at 37 °C.

### 2.3. Transfection and RNA Interference

Mouse KLK6 cDNA (NM_001164696) purchased from Origene (Rockville, MD, USA) was used for protein transfection. The KLK6 expression vector was transfected using lipofectamine as the manufacturer’s protocol. Briefly, cells that were at 5 × 10^5^ cells/well in 6-well plates were washed twice with DMEM without FBS and were mock-transfected or transfected with KLK6; then, the cells were incubated for 12–24 h at 37 °C, following which the transfected cells were used for the subsequent experiments. Mouse KLK6 cDNA (NM_001164696) was purchased from Origene (Rockville, MD, USA). The KLK6 expression vector was transfected using lipofectamine as per the manufacturer’s protocol. siRNA duplexes of KLK6, PAR1, PAR2, and control (non-targeting) were obtained from Bioneer (Daedeok-gu, Daejeon, Korea). siRNA duplexes were transfected with the lipofectamine RNAiMAX reagent according to the manufacturer’s protocol. All siRNA sequences are listed in Appendix A.

### 2.4. Generation of KLK6-Deficient Mice

A single-guide RNA (sgRNA) (5′-CAGGCTGCCCTCTACACCTCAGG-3′) targeting exon 4 of murine KLK6 was designed and microinjected along with Cas9 mRNA into 4- to 6-week-old female embryos of C57BL/6J mice (Appendix A). Chimeric offspring were backcrossed to C57BL/6J mice, and germline transmission was confirmed by polymerase chain reaction (PCR) and enzyme cutting of tail genomic DNA. PCR genotyping of KLK6 tail genomic DNA was performed using the primers 5′-CCC TCTACACCTCAGGTCACTTG-3′ and 5′-TCCTCAGAGCAGTCATTCTTCAA-3′, which amplified a 324-bp targeted band. The restriction enzyme site was the KLK6 deletion site; to confirm wild-type, heterozygous, and homozygous KLK6, Bsu36I (New England Biolabs, Ipswich, MA, USA) was used for enzyme cutting (Appendix A). In the case of the WT mice, cutting at the Bsu36I enzyme site yielded two bands, that is, 210- and 114-bp bands, whereas one 324-bp band was obtained for the KLK6-deficient mice because of the deletion of the Bsu36I enzyme site (Appendix A). Deletion of nucleotides 141–150 in the KLK6 sequence of KLK6-deficient mice was confirmed by sequencing analysis (Appendix A). Furthermore, KLK6 expression was detected in the serum of WT mice but not in the serum of KLK6-deficient mice (Appendix A).

### 2.5. Animal Experiments

C57BL/6 mice (age, 6–10 weeks) used in animal experiments were purchased from the Korea Research Institute of Bioscience and Biotechnology (KRIBB; Cheongju, Ochang, Republic of Korea). B16F10 cells or LLC cells (1 × 10^6^ in 100 µL phosphate-buffered saline) were subcutaneously injected into mice. The tumor volume was measured every other day after subcutaneous injection. Mice were sacrificed when the largest tumor volume reached 2000 mm^3^. The tumor volume was calculated based on the following formula: 0.52 × (major axis) × (minor axis) × (height) after subcutaneous injection of B16F10 cells or LLC cells (1 × 10^6^ in 100 µL phosphate-buffered saline). B16F10 cells (5 × 10^5^ cells per mouse) or LLC cells (1 × 10^6^ cells per mouse) suspended in 100 µL PBS were intravenously injected into the tail vein for lung metastasis analysis. The mice were sacrificed two weeks or four weeks later, respectively. The mice were anesthetized via an intraperitoneal injection of avertin (500 mg/kg). After anesthesia, blood samples were collected by cardiac puncture of the mice.

To study the effect of CXCL1 neutralization and PAR1 inhibition, B16F10 and LLC cells were subcutaneously or intravenously injected in mice. After 1 h, mice were injected with PBS or CXCL1 neutralizing antibody (5 mg/kg, MAB453) every other day for 10 days. Neutralization of CXCL1 followed the optimal dose as previously reported [31,32]. Likewise, B16F10 and LLC were subcutaneously or intravenously injected in mice. After 1 h, mice were injected with DMSO or PAR1 antagonist (5 mM/kg, SCH 79797) on every other day for 10 days. PAR1 antagonist followed the optimal dose as previously reported [33,34]. All animal studies were performed by the guidelines of and with the permission of the KRIBB Institutional Animal Care and Use Committee (KRIBB-AEC-18170).

### 2.6. Bone Marrow–Derived Macrophage (BMDM) Isolation

Bone marrow was isolated from the femurs of 6- to 10-week-old WT and KLK6-deficient mice. Fresh bone marrow cells were plated on DMEM plus 10% FBS, supplemented with 30% L929-conditioned medium as a source of GM-CSF. Fresh medium was added every three days, and BMDMs were harvested after 7 days of culture. Harvested BMDMs were washed three times with PBS and used for the subsequent experiment.

### 2.7. In Vivo Macrophage Depletion

B16F10 cells (1 × 10^6^ in 100 µL PBS) were subcutaneously injected into WT and KLK6−/− mice. A macrophage depletion kit containing control liposomes and clodronate liposomes (Encapsula NanoSciences, Brentwoood, TN, USA) was intraperitoneally injected every five days at a 200 uL per mouse dose.

### 2.8. Coculture Assay and Conditioned Medium (CM) Production

WT and KLK6-deficient BMDMs (2 × 10^6^ cells/well) were seeded into a 6-well plate, and B16F10 and LLC cells (5 × 10^5^ cells/well) were seeded into an insert plate for 24 h (pore size, 0.4 μm; SPL, Pocheon-si, Gyeonggi-do, Korea). For obtaining CM, WT, and KLK6-deficient BMDMs were cultured in 100-mm cell culture dishes at 37 °C in a humidified atmosphere of 5% CO_2_ in air for 24 h.

### 2.9. Reverse Transcriptase-Polymerase Chain Reaction (RT-PCR)

The total RNA from B16F10 cells, LLC cells, and macrophages was extracted using TRIzol (Invitrogen) following the manufacturer’s recommendations. CXCL1, CXCL2, PAR1, PAR2, KLK6, and glyceraldehyde-3-phosphate dehydrogenase (GAPDH) expression levels were measured by RT-PCR analysis using cDNA synthesized from 5 µg total RNA with reverse transcription kits (Promega, Madison, WI, USA). One microliter of cDNA was used for the PCR, and duplicate reactions were performed for each sample. GAPDH was used to normalize the level of mRNA expression. The sequences of the primers used in this study are listed in Appendix A.

### 2.10. Western Blot Analysis

The cells were solubilized in radioimmunoprecipitation assay (RIPA) lysis buffer. Protein lysates were quantified using the Pierce^®^BCA Protein Assay Kit (Thermo Scientific, Rockford, IL, USA). Equal amounts of protein samples were separated via 6–15% SDS-PAGE and transferred onto PVDF membranes using a Trans-Blot^®^Turbo™Transfer pack (Bio-Rad, Hercules, CA, USA). The membranes were blocked with 5% skimmed milk in TBS-T for 2 h and incubated overnight at 4 °C with primary antibodies. The bound antibodies were visualized with horseradish peroxidase-conjugated secondary antibodies. The membranes were developed with enhanced chemiluminescence reaction according to the manufacturer’s instructions.

### 2.11. Cytokine Array and Enzyme-Linked Immunosorbent Assay (ELISA)

A mouse cytokine array panel A (ARY006, R&D Systems) was used for the mouse cytokine array according to the manufacturer’s instructions. The levels of CXCL1, CXCL2, TNF-α, and IFN-γ were measured using the mouse CXCL1 ELISA kit (LS-F268, LSBio, Seattle, Washington, USA), mouse CXCL2 ELISA kit (LS-F621, LSBio), mouse TNF-α DuoSet ELISA kit (DY-410, R&D Systems), and mouse IFN-γ DuoSet ELISA kit (DY-485, R&D Systems), respectively. All ELISAs were performed according to the manufacturers’ instructions.

### 2.12. Flow Cytometry Analysis

Isolated cells in the lung were stained with indicated antibodies for further analysis. For immunostaining, the cells were washed two times with PBS containing 2% FBS, adjusted to approximately 1 × 10^5^ to 1 × 10^6^ cells in 100 μL of the same buffer, and labeled with PE-anti-F4/80 (BD Bioscience, 565410), PE-Cy7-anti-CD11b (BD Bioscience, 561098), and FITC-anti-TNF-α (BD Bioscience, 554418). Incubations with antibodies were performed for 30 min at 4 °C in the dark, and then cells were centrifuged for 4 min at 900 rpm. After removal of the supernatant, the cell pellet was washed once with PBS containing 2% FBS and then resuspended in fixation and permeabilization buffer (BD Bioscience, Franklin Lakes, NJ, USA) for 30 min at 4 °C. Cells were washed two times with PBS containing 2% FBS and then acquired by FACSVerse (BD Bioscience) and analyzed using software Flow Jo (Tree Star, Inc., Ashland, OR, USA).

### 2.13. Statistical Analysis

All quantitative data are expressed using the mean ± standard deviation (SD); in vivo data are expressed as mean ± standard error of the mean (SEM). Statistical analysis was performed using SAS 9.2 software (SAS Institute, Cary, NC, USA). When the *p*-value was < 0.05, the differences between groups were statistically significant.

## 3. Results

### 3.1. KLK6 Promoted Tumor Growth and Metastasis

We generated KLK6−/− mice to investigate the role of KLK6 in the tumor microenvironment. These mice showed no obvious phenotype during embryonic development and had a normal life span. We subcutaneously injected B16F10 melanoma or LLC lung cancer cells into WT and KLK6−/− mice to investigate the effects of KLK6 on tumor growth in vivo. KLK6−/− mice injected with B16F10 or LLC cells showed significantly less tumor volume and weight than the WT mice (Figure 1A–F). Western blot analysis revealed that Ki67 and VEGF1 protein levels were decreased, and cleaved caspase-3 protein levels were increased in tumor tissues from KLK6−/− mice injected with B16F10 cells compared to WT mice (Appendix A). This result suggests that KLK6 may increase the proliferation and reduce the apoptosis of cancer cells in vivo. To assess the effects of KLK6 on tumor metastasis, we injected B16F10 or LLC cancer cells into WT and KLK6−/− mice via the tail vein. Numerous metastatic lung nodules were observed in WT mice injected with B16F10 or LLC cells, whereas a considerably lower number of nodules was detected in KLK6−/− mice injected with cancer cells (Figure 1G–J). These results indicate that KLK6 plays a prominent role in tumor growth and metastasis in vivo.

To identify the secreted factors that regulate KLK6-mediated malignant progression, we used mouse cytokine arrays for serum samples from WT and KLK6−/− mice injected with B16F10 melanoma cells. Among the 24 cytokines, we found that serum CXCL1 and CXCL2 levels were significantly higher in WT mice injected with B16F10 cells than in KLK6−/− mice injected with B16F10 cells (Figure 2A). We also confirmed CXCL1 and CXCL2 secretion by using ELISA; the serum CXCL1 levels of WT mice were significantly higher than those of KLK6−/− mice, whereas the CXCL2 level was slightly increased (Figure 2B,C).

To investigate whether CXCL1 can affect melanoma growth and metastasis, we subcutaneously or intravenously injected WT mice with B16F10 melanoma cells and then administered CXCL1 neutralizing antibody every other day for 10 days. CXCL1 inhibition significantly attenuated tumor growth (Figure 2D,E) and metastatic lung nodule formation (Figure 2F,G). Moreover, CXCL1 neutralizing antibody reduced tumor growth and metastasis of LLC lung carcinoma in vivo (Appendix A). These findings suggest that CXCL1 is required for tumor growth and metastasis.

### 3.2. KLK6 Stimulated CXCL1 Expression in Cancer Cells by Promoting TNF-α Secretion in Macrophages

In the tumor microenvironment, macrophages and other stromal cells promote CXCL1 expression in cancer cells to promote cancer survival at metastatic sites [35,36]. Because KLK6 expression is detected in macrophages [37], we hypothesized that macrophages might affect KLK6-mediated tumor growth. To test this possibility, we depleted macrophages from tumor xenografts by the administration of clodronate liposome. Macrophage depletion suppressed tumor growth in WT mice. However, this inhibitory effect was reduced in KLK6−/− mice (Appendix A), suggesting that macrophages are, at least partially, important for KLK6-mediated tumor growth.

Next, we isolated BMDMs from WT or KLK6−/− mice to investigate whether macrophages affect KLK6-mediated CXCL1 expression in cancer cells. Western blot analysis showed that KLK6 protein was present in the CM of WT BMDMs but not in that of KLK6−/− BMDMs (Figure 3A). Next, we treated with the CM of WT or KLK6−/− BMDM into B16F10 melanoma cells. CXCL1 mRNA expression in B16F10 cells significantly increased on treatment with the CM of WT BMDMs. However, this induction was markedly lower in cancer cells treated with the CM of KLK6−/− BMDMs (Figure 3B). CXCL2 mRNA expression increased on treatment with the CM of WT BMDMs or KLK6−/− BMDMs (Figure 3B). We performed ELISA to determine whether macrophages affect CXCL1 and CXCL2 secretion in cancer cells. We found that the secreted CXCL1 levels in B16F10 cells treated with CM of KLK6−/− BMDMs were significantly lower than those treated with WT BMDMs (Figure 3C). Next, we confirmed the effect of macrophages on CXCL1 production in cancer cells using a coculture system. KLK6 mRNA expression was detected in WT BMDMs but not in KLK6−/− BMDMs (Figure 3D). RT-PCR and ELISA data showed that CXCL1 mRNA levels and secretion in B16F10 melanoma significantly increased on coculture with WT BMDMs but not with KLK6−/− BMDMs (Figure 3E,F). Moreover, CXCL1 mRNA expression was not detected in WT and KLK6−/− BMDMs on coculture with cancer cells. However, when cocultured, CXCL2 mRNA expression increased in the WT/KLK6−/− BMDMs and cancer cells (Appendix A). These results indicate that KLK6 stimulated CXCL1 expression in cancer cells when cocultured with BMDMs, whereas it does not affect CXCL2 expression in cancer cells and BMDMs.

To confirm the effect of KLK6 on macrophage-induced CXCL1 production, RAW 264.7 mouse macrophage cells were transfected with GFP-tagged KLK6. Western blot data showed that KLK6 protein was expressed in GFP-tagged KLK6-transfected cells but not in the control vector-transfected cells (Figure 4A). We treated B16F10 cells with the CM of the control vector- or KLK6-transfected RAW 264.7 cells, following which CXCL1 production was analyzed by RT-PCR and ELISA. CXCL1 mRNA levels and secretion increased in B16F10 melanoma treated with the CM of KLK6 overexpressed-RAW264.7 cells= compared to the CM of control vector-transfected cells (Figure 4B,C). To further confirm this effect, we depleted endogenous KLK6 in the RAW 264.7 mouse macrophage cell line using siRNA. Cells transfected with the two different KLK6 siRNAs had significantly lower KLK6 mRNA levels than cells transfected with control siRNA (Figure 4D). We treated with the CM of the control or that of RAW 264.7 cells transfected with the two KLK6 siRNAs into cancer cells. CXCL1 production was analyzed by RT-PCR and ELISA. Consistent with the results for WT and KLK6−/− BMDMs, the CXCL1 mRNA levels and secretion in B16F10 cells treated with the CM of KLK6 siRNA transfected–RAW 264.7 cells were significantly lower than the levels in those treated with the CM of control siRNA transfected–RAW 264.7 cells. However, KLK6 silencing did not affect macrophage-induced CXCL2 production in cancer cells (Figure 4E,F). These results indicate that KLK6 is responsible for macrophage-promoted CXCL1 production in cancer cells.

Macrophages can produce several proinflammatory cytokines including TNF-α, IL-1β, and IFN-γ [38]. In addition, previous studies have suggested that these cytokines promote CXCL1 expression in several cancers [39,40,41]. To confirm whether these cytokines affect CXCL1 expression, we treated B16F10 and LLC cancer cells with TNF-α, IL-1β, or IFN-γ. RT-PCR and ELISA analyses showed that TNF-α and IFN-γ significantly increased CXCL1 production, whereas IL-1β only slightly increased it (Figure 5A,B). We measured the TNF-α and IFN-γ levels in the supernatants of WT and KLK6−/− BMDMs after being co-cultured with B16F10 melanoma cells. WT BMDMs showed higher TNF-α levels than KLK6−/− BMDMs (Figure 5C); however, the IFN-γ levels did not significantly differ between the two types of BMDMs (Figure 5D). Moreover, KLK6 overexpression increased TNF-α secretion in the RAW 264.7 cells, whereas KLK6 silencing decreased them (Figure 5E,F). These results suggest that KLK6 promotes TNF-α production in an autocrine manner.

To determine the effect of TNF-α on macrophage-mediated CXCL1 production in cancer cells, we treated B16F10 cells with the CM of mock-transfected or KLK6-transfected RAW 264.7 cells in the presence or absence of the TNF-α neutralizing antibody. RT-PCR and ELISA data showed that B16F10 cells treated with the CM of KLK6-overexpressing RAW 264.7 cells had higher CXCL1 mRNA and protein levels than the mock-transfected cells; however, this enhancement was suppressed by the TNF-α neutralizing antibody (Figure 5G,H). In addition, Western blot analysis showed that TNF-α and CXCL1 protein were detected in tumor tissues from WT mice injected with B16F10 melanoma but not in those from KLK6−/− mice injected with B16F10 cells (Appendix A). These results indicate that KLK6 promoted TNF-α production by macrophages, leading to CXCL1 production in cancer cells.

### 3.3. KLK6 Promoted TNF-α Production in Macrophages via PAR1

Previous studies suggested that KLK6 transduced intracellular signaling through PARs in an autocrine or paracrine manner [13]. Therefore, we assessed whether PARs regulated KLK6-mediated production of TNF-α. We transfected the control siRNA, two PAR1 siRNAs, or two PAR2 siRNAs into RAW 264.7 cells. Western blot analysis showed that PAR1 and PAR2 siRNAs significantly reduced the PAR1 and PAR2 protein levels, respectively (Figure 6A). The level of the TNF-α protein in the supernatants of RAW 264.7 cells was markedly suppressed by PAR1 depletion but not PAR2 depletion (Figure 6B), suggesting that PAR1 is required for TNF-α production in macrophages. Next, we determined whether PAR1 depletion in macrophages affected CXCL1 production in cancer cells. To test this, we treated B16F10 and LLC cancer cells with the CM of RAW 264.7 cells transfected with the control siRNA, PAR1 siRNAs, or PAR2 siRNAs. Treatment with the CM of control siRNA–transfected RAW 264.7 cells significantly increased CXCL1 levels in the B16F10 and LLC cancer cell lines; this enhancement was suppressed in cancer cells treated with the CM of PAR1 siRNA–transfected RAW 264.7 cells but not in those treated with the CM of control siRNA or PAR2 siRNAs-transfected RAW 264.7 cells (Figure 6C). Moreover, ELISA showed that treatment with PAR1 siRNAs led to lower levels of CXCL1 secretion than those obtained with the control and PAR2 siRNAs (Figure 6D).

To further confirm the effect of PARs on the production of TNF-α in macrophages and CXCL1 in cancer cells, we treated RAW 264.7 cells with a PAR1 or PAR2 inhibitor. Consistent with the siRNA results, the PAR1 inhibitor suppressed TNF-α production in the RAW 264.7 cells and macrophage-mediated CXCL1 production in B16F10 and LLC cancer cells. However, treatment with PAR2 inhibitor did not affect CXCL2 production (Figure 6E–G). These results indicate that PAR1 activity is necessary for TNF-α production in macrophages, which promotes CXCL1 expression in cancer cells.

We then assessed whether PAR1 signaling is involved in KLK6-mediated TNF-α production. Recombinant murine KLK6 protein (rKLK6) was generated and used to treat KLK6−/− BMDMs. rKLK6 increased the TNF-α levels in the supernatant of KLK6−/− BMDMs in a concentration-dependent manner (Figure 7A), suggesting that rKLK6 can promote TNF-α production in macrophages. The CM of KLK6−/− BMDMs treated with rKLK6 caused the increase in CXCL1 mRNA expression and secretion in a dose-dependent manner (Figure 7B,C). The CM of KLK6−/− BMDMs treated with 10 µg/mL rKLK6 increased the level of CXCL1 secreted from B16F10 cells by the amount of the CM of WT BMDMs (Figure 7B,C). Therefore, we used 10 µg/mL rKLK6 for further experiments. To verify the effect of the PAR1 inhibitor on KLK6-mediated TNF-α production, we treated the cells with 10 µg/mL rKLK6 in the presence or absence of the PAR1 inhibitor, and the TNF-α level in the supernatant was measured by ELISA. Treatment with the PAR1 inhibitor suppressed rKLK6-mediated TNF-α production (Figure 7D); moreover, it reduced CXCL1 production in B16F10 cells treated with the CM of KLK6−/− BMDMs treated with rKLK6 (Figure 7E). These results indicate that PAR1 signaling mediates KLK6-dependent TNF-α production in macrophages.

### 3.4. PAR1 Inhibitor Suppressed rKLK6-Mediated Tumor Growth and Lung Metastasis

Because our results established that KLK6/PAR1 signaling mediates the crosstalk between macrophages and cancer cells, we assessed the effect of PAR1 on KLK6-mediated tumor growth in vivo. For this analysis, we subcutaneously injected B16F10 melanoma cells into KLK6−/− mice, which showed low tumor growth (Figure 8). After 1 h, the mice were injected with DMSO, rKLK6 alone, or rKLK6 along with the PAR1 inhibitor every other day for 10 days. Compared to the DMSO-injected mice, rKLK6-injected mice showed enhanced tumor growth (Figure 8A–C), suggesting that rKLK6 can promote tumor growth in vivo. rKLK6 also increased TNF-α and CXCL1 levels in mouse serum (Figure 8D,E). However, the PAR1 inhibitor reduced the rKLK6-promoted tumor growth and serum levels of TNF-α and CXCL1 (Figure 8A–E). Moreover, Western blot analysis showed that the CXCL1 protein was detected in tumor tissue from mice injected with rKLK6 only, but not in those from mice injected with the rKLK6/PAR1 inhibitor (Appendix A).

To verify the effects of PAR1 on rKLK6-mediated lung metastasis, we injected KLK6−/− mice with B16F10 cells via the tail vein. After 1 h, the mice were injected with PBS, rKLK6 alone, or rKLK6 along with the PAR1 inhibitor. The mice injected with rKLK6 showed numerous lung metastatic nodules (Figure 8F–H and Appendix A) and enhanced serum TNF-α and CXCL1 levels (Figure 8I,J). However, this enhancement was suppressed by the PAR1 inhibitor (Figure 8F–J and Appendix A). CXCL1 inhibition by neutralizing antibodies also attenuated rKLK6-promoted tumor growth and lung metastasis (Appendix A). These findings indicate that KLK6/PAR1 signaling mediates TNF-α and CXCL1 production and drives tumor growth and lung metastasis in vivo.

## 4. Discussion

Numerous studies have suggested that KLK6 is aberrantly expressed in and is associated with malignant progression in various human cancers including ovarian, colon, lung, and breast cancers [22,23,24,25,26,27]. Serum KLK6 levels increased in patients with metastatic breast cancer and colon cancer [22,39]. Recent research has shown that KLK6−/− mice are highly resistant to skin tumor growth and development and that this resistance may be associated with reduced TPA-induced inflammation [30]. KLK6 expression is upregulated during colon cancer progression; high expression is observed in inflammatory cells in colorectal adenocarcinoma [42]. Moreover, multiple studies have suggested that KLK6 is associated with various inflammatory diseases including psoriasis, atopic dermatitis, and inflammatory joint disease [16,17,18,19,20]. These studies suggest that KLK6 could be correlated with tumor inflammation.

In the current study, we provide experimental evidence indicating a potential role played by KLK6 in cross-talk between melanoma cells and macrophages in the tumor microenvironment. Our coculture system data showed lower TNF-α levels in KLK6−/− BMDMs than in the WT BMDMs. Moreover, the TNF-α protein levels were increased in macrophages isolated from the tumor tissues of WT mice injected with B16F10 cells compared to KLK6−/− mice (Appendix A). The mice injected with rKLK6 alone showed numerous lung metastatic nodules and enhanced serum TNF-α levels. However, PAR1 inhibition suppressed rKLK6-mediated TNF-α production in KLK6−/− mice. Although we cannot exclude the possibility of the effect of other microenvironmental factors and inflammatory cytokines on TNF-α production in a tumor microenvironment, these results suggest that the KLK6/PAR1 axis is critical, at least in part, for TNF-α production by macrophage.

TNF-α secreted by macrophages stimulated CXCL1 production in melanoma cells to facilitate tumor growth and metastasis. CXCL1 is known to be involved in cancer development and malignant progression including tumor growth, metastasis, and chemoresistance [43]. Our experiment using CM and the coculture system showed that KLK6 mediated TNF-α production in macrophages in an autocrine manner. The treatment with the TNF-α neutralizing antibody in the CM of WT BMDMs and KLK6-overexpressing RAW 264.7 cells suppressed CXCL1 production from cancer cells. These results suggest that TNF-α is critical for CXCL1 production from cancer cells. In line with these findings, a recent study reported that TNF-α released from endothelial cells stimulated CXCL1 expression in LM2 tumor cells via nuclear factor (NF)-κB activation [44]. Macrophages produce many pro-inflammatory cytokines including TNF-α, IL-1β, and IFN-γ, which can promote CXCL1 expression [35,36,37]. Therefore, we also measured the IFN-γ levels in the supernatants of WT and KLK6−/− BMDMs. However, the IFN-γ levels did not significantly differ between the two BMDMs, suggesting that IFN-γ is not involved in KLK-mediated CXCL1 expression. In the current study, inhibition of CXCL1 using a neutralizing antibody suppressed tumor growth and lung metastasis in vivo. Moreover, CXCL1 inhibition rKLK6-mediated tumor growth (Appendix A) and metastatic lung module formation (Appendix A) in KLK6−/− mice injected with B16F10. Therefore, although we cannot exclude the possibility that KLK6 is associated with other mediators such as CXCL2, its promoting effect on malignant progression is, at least partially, attributable to the production of CXCL1.

KLK6, a serine protease, regulates various biological processes via receptor-dependent and receptor-independent mechanisms. KLK6 mediates E-cadherin shedding to promote keratinocyte migration and invasion [28]. It can also activate PARs to transduce intracellular signaling in an autocrine or paracrine manner. KLK6 promotes cerebellar neuron degeneration and exacerbates glutamate neurotoxicity via PAR1 and PAR2 [21]. However, KLK6 overexpression causes psoriasis and inflammatory arthritis via signaling through PAR1. In the current study, we showed the inhibition of PAR1 via RNA interference and that a specific inhibitor suppressed TNF-α expression stimulated by KLK6 in macrophages, which suggests that PAR1 plays a significant role in KLK6-mediated TNF-α production. PAR1 expression has been detected in cancer-related fibroblasts, blood vessel myocytes, mast cells, and macrophages in the tumor microenvironment [45]. PAR1 signaling stimulates the expression of various growth factors in macrophages and is also involved in psoriasis via the activation of STAT3, which can stimulate TNF-α, IL-23, and IL-17 production [20,46]. Therefore, STAT3 may be a downstream mediator of KLK6/PAR1-mediated TNF-α production. Several studies have suggested that PAR1 is associated with acquiring malignant phenotypes via promoting cancer cell migration and invasion [47]. Therefore, PAR1 is considered a therapeutic target for cancer treatment. In line with this, our results showed that the PAR1 inhibitor suppressed TNF-α and CXCL1 production and rKLK6-promoted tumor growth and metastasis in the xenograft model, suggesting that PAR1 plays a critical role in KLK6-mediated malignant progression.

## 5. Conclusions

We provide experimental evidence indicating a potential role played by KLK6 in cross-talk between melanoma cells and macrophages in the tumor microenvironment. KLK6 promoted the production of TNF-α from macrophages via PAR1. TNF-α secreted by macrophages stimulated CXCL1 production in melanoma cells to facilitate tumor growth and metastasis. Therefore, our results suggest that inhibition of the KLK6/PAR1 axis would provide a potential therapeutic strategy for cancer.

## Figures and Tables

**Figure 1 cells-11-04101-f001:**
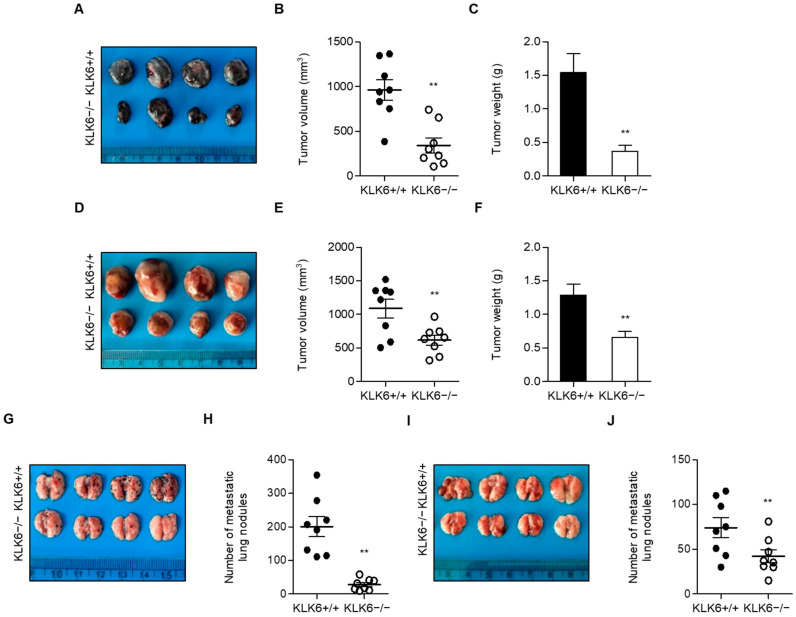
KLK6 deficiency attenuated tumor growth and metastasis in vivo. 1 × 10^6^ cells of B16F10 (**A**–**C**) or LLC (**D**–**F**) cells were subcutaneously injected into WT and KLK6−/− mice (*n* = 8 per group). (**A**,**D**) Representative photograph of tumors. Tumor volume (**B**,**E**) and tumor weight (**C**,**F**) were calculated as described in the Materials and Methods. 5 × 10^5^ cells of B16F10 (**G**,**H**) or LLC (**I**,**J**) cells were intravenously injected into the tail vein of WT or KLK6−/− mice (*n* = 8 per group). (**G**,**I**) Representative photograph of metastatic lung nodules from WT and KLK6−/− mice. (**H**,**J**) The number of metastatic lung nodules was calculated as described in the Materials and Methods. ** *p* < 0.01 (Student’s *t*-test). Data are representative of three experiments. +/+, wild-type; −/−, homozygous knockout.

**Figure 2 cells-11-04101-f002:**
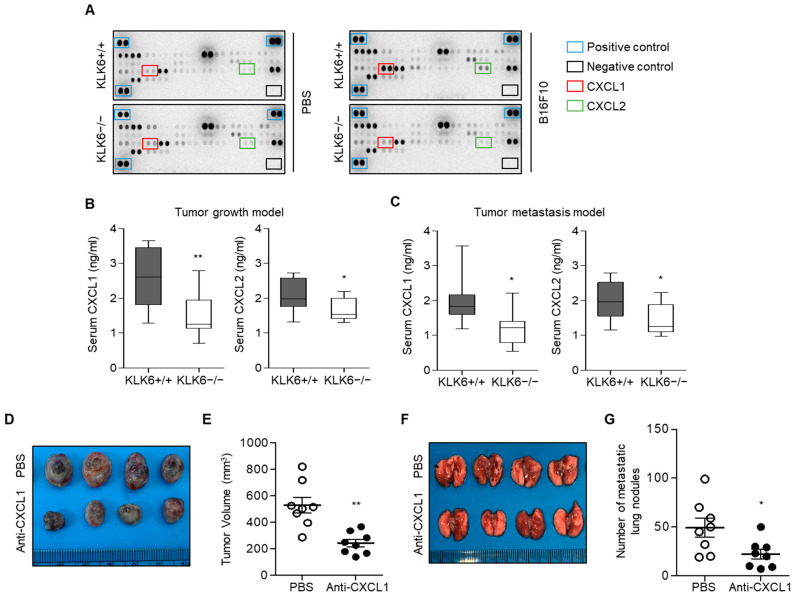
KLK6-mediated CXCL1 production was required for tumor growth and metastasis. (**A**) Mouse cytokine array in serum from WT or KLK6−/− mice injected with PBS (control) or B16F10 melanoma cells. (**B**,**C**) 1 × 10^6^ of B16F10 cells were subcutaneously (**B**) or intravenously (**C**) injected into WT and KLK6−/− mice (*n* = 8 per group). CXCL1 and CXCL2 levels in serum from WT and KLK6−/− mice injected with B16F10 melanoma cells were measured by ELISA as described in the Materials and Methods. (**D**,**E**) 1 × 10^6^ of B16F10 cells were subcutaneously injected into WT mice (*n* = 6 per group). After 1 h, mice were injected with PBS (control) or CXCL1 neutralizing antibody (5 mg/kg) every other day for 10 days. (**D**) Representative photograph of the tumors. (**E**) Tumor volume was measured as described in the Materials and Methods. (**F**,**G**) 5 × 10^5^ of B16F10 cells were intravenously injected into the tail vein of WT mice (*n* = 8 per group). After 1 h, the mice were injected with PBS (control) or CXCL1 neutralizing antibody (5 mg/kg) every other day for 10 days. (**F**) Representative photograph of metastatic lung nodules. (**G**) The number of metastatic lung nodules was calculated as described in the Materials and Methods. ** *p* < 0.01 and * *p* < 0.05 (Student’s *t*-test). Data are representative of three experiments.

**Figure 3 cells-11-04101-f003:**
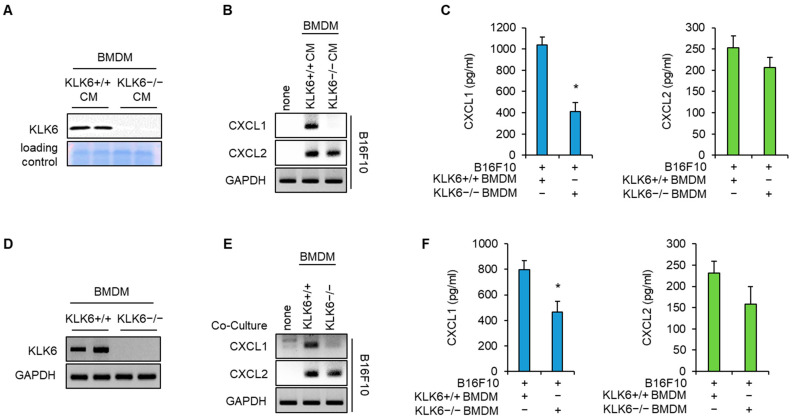
KLK6 deficiency abolished BMDM-mediated CXCL1 production in cancer cells. Conditioned medium (CM) was collected from WT and KLK6−/− BMDMs. (**A**) KLK6 protein expression in CM of WT and KLK6−/− BMDMs were analyzed by Western blot. (**B**,**C**) B16F10 cells were treated with the CMs of WT and KLK6−/− BMDMs. mRNA (**B**) and protein (**C**) levels of CXCL1 and CXCL2 were analyzed by RT-PCR and ELISA, respectively. (**D**) KLK6 mRNA levels in WT and KLK6−/− BMDMs were analyzed by RT-PCR. (**E**,**F**) B16F10 cells were cocultured with WT or KLK6−/− BMDMs. mRNA (**E**) and protein (**F**) levels of CXCL1 and CXCL2 were analyzed by RT-PCR and ELISA, respectively. * *p* < 0.05 (Student’s *t*-test). Data are representative of three experiments. +/+, wild-type; −/−, homozygous knockout.

**Figure 4 cells-11-04101-f004:**
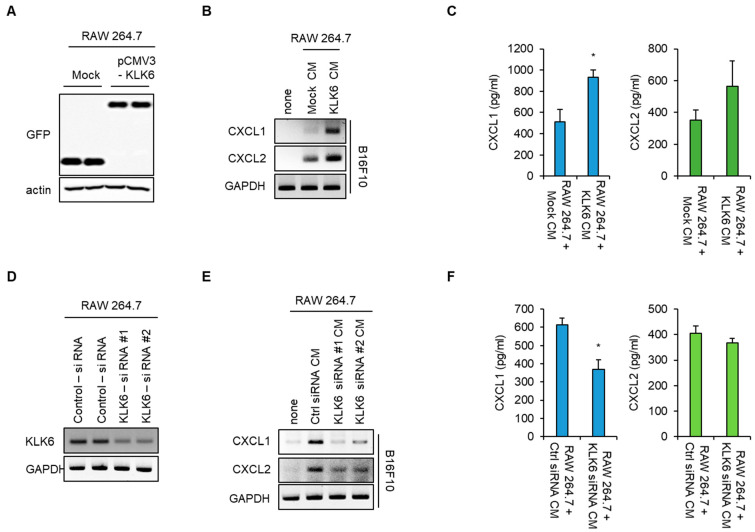
KLK6 is responsible for macrophage-promoted CXCL1 production in cancer cells. (**A**) RAW 264.7 cells were transfected with the mock vector (Mock) or GFP-tagged KLK6 expression vector (pCMV3-KLK6). KLK6 protein expression in CM of mock- or pCMV3-KLK6-transfected cells was analyzed by Western blot. (**B**,**C**) B16F10 cells were treated with the CMs of mock- or pCMV3-KLK6-transfected cells for 24 h. (**B**) The mRNA levels of the CXCL1 and CXCL2 in the indicated CM-treated B16F10 cells were analyzed by RT-PCR. (**C**) The protein levels of CXCL1 and CXCL2 in the supernatants of the indicated CM-treated B16F10 cells were analyzed by ELISA. (**D**) RAW 264.7 cells were transfected with the control or two different KLK6 siRNAs. KLK6 mRNA levels were analyzed by RT-PCR. The KLK6 secreted in CMs was analyzed by Western blot (right panel). (**E**,**F**) B16F10 cells were treated with the CMs of RAW 264.7 cells transfected with the indicated siRNA for 24 h. (**E**) The mRNA levels of CXCL1 and CXCL2 in the indicated CM-treated B16F10 cells were analyzed RT-PCR. (**F**) The protein levels of CXCL1 and CXCL2 in the supernatants of the indicated CM-treated B16F10 cells were analyzed by ELISA. Statistical significance was determined by the Student’s *t*-test. * *p* < 0.05. Data are representative of three experiments.

**Figure 5 cells-11-04101-f005:**
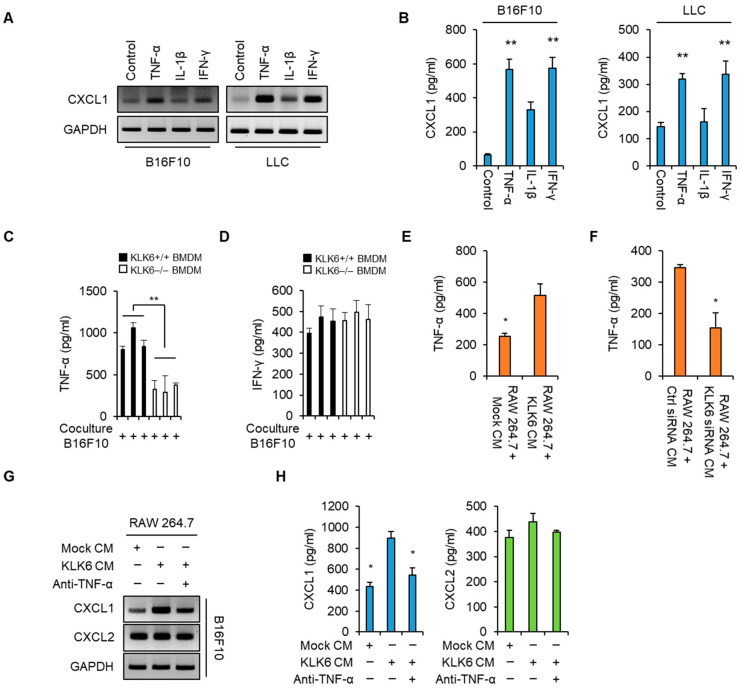
KLK6 promoted CXCL1 production by stimulating TNF-α secretion by macrophages. (**A**,**B**) B16F10 and LLC cells were treated with TNF-α (100 ng/mL), IL-1β (100 ng/mL), or IFN-γ (100 ng/mL) for 24 h. (**A**) The mRNA levels of CXCL1 in the indicated cytokine-treated B16F10 and LLC cells were analyzed by RT-PCR. (**B**) The protein levels of CXCL1 in the supernatants of the indicated cytokine-treated B16F10 and LLC cells were analyzed by ELISA. The levels of TNF-α (**C**) and IFN-γ (**D**) in the supernatants of WT or KLK6−/− BMDMs were analyzed by ELISA. (**E**) The secreted TNF-α levels in the supernatants of the mock- or pCMV3-KLK6-transfected RAW 264.7 cells were measured by ELISA. (**F**) The secreted TNF-α levels in the supernatants of the control siRNA- or KLK6 siRNA-transfected RAW 264.7 cells were measured by ELISA. (**G**,**H**) B16F10 cells were treated with the CMs of RAW 264.7 cells that were the mock vector or KLK6-expressing vector in the presence or absence of the TNF-α neutralizing antibody (2 µg/mL) for 24 h. (**G**) The mRNA levels of CXCL1 and CXCL2 in the indicated CM-treated B16F10 cells were analyzed by RT-PCR. (**H**) The protein levels of CXCL1 and CXCL2 in the supernatants of the indicated CM-treated B16F10 cells were analyzed by ELISA. Statistical significance was determined by the Student’s *t*-test. * *p* < 0.05 and ** *p* < 0.01. Data are representative of three experiments.

**Figure 6 cells-11-04101-f006:**
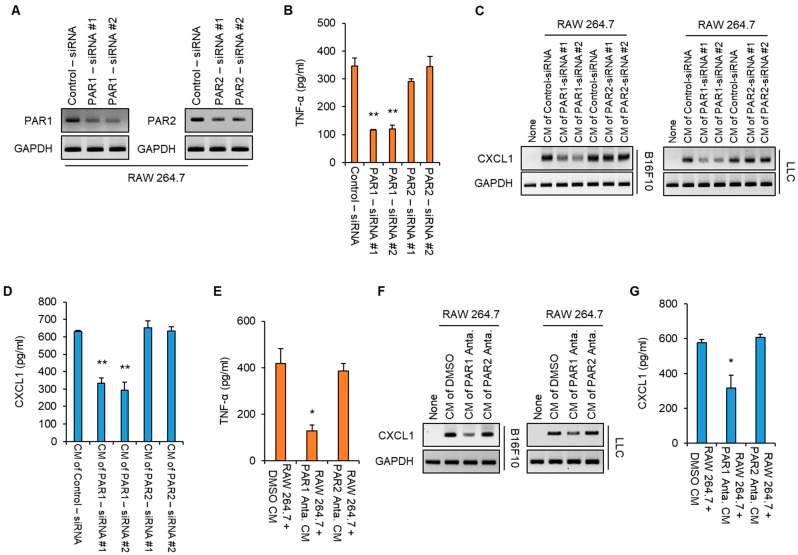
KLK6 stimulated TNF-α production in macrophages via PAR1. (**A**–**D**) RAW 264.7 cells were transfected with control siRNA, two PAR1 siRNAs, or two PAR2 siRNAs. (**A**) PAR1 and PAR2 mRNA levels in the indicated siRNAs-transfected cells were measured by RT-PCR. (**B**) The level of secreted TNF-α in the supernatants of the indicated siRNAs-transfected cells was analyzed by ELISA. (**C**,**D**) B16F10 and LLC cells were treated with the CMs of the indicated siRNA-transfected RAW 264.7 cells for 24 h. The mRNA (**C**) and secreted protein (**D**) of CXCL1 in B16F10 and LLC cells treated with the CMs of indicated RAW 264.7 cells were analyzed by RT-PCR and ELISA, respectively. (**E**–**G**) RAW 264.7 cells were treated with DMSO (as control), a PAR1 antagonist, or a PAR2 antagonist for 24 h. (**E**) The level of secreted TNF-α in the supernatants of the indicated RAW 264.7 cells was analyzed by ELISA. (**F**) The level of CXCL1 mRNA in the B16F10 and LLC cells treated with the CMs of indicated RAW 264.7 cells was analyzed by RT-PCR. (**G**) The level of secreted CXCL1 in the supernatants of B16F10 cells treated with the CMs of the indicated RAW 264.7 cells was measured by ELISA. Statistical significance was determined by the Student’s *t*-test. * *p* < 0.05 and ** *p* < 0.01. Data are representative of three experiments.

**Figure 7 cells-11-04101-f007:**
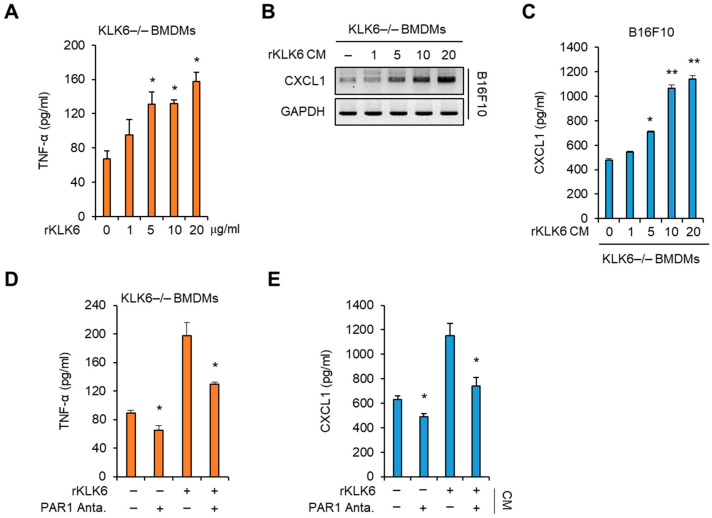
Recombinant KLK6 promoted macrophage-derived TNF-α production via PAR1 in KLK6−/− BMDMs. (**A**) KLK6−/− BMDMs were treated with rKLK6 at the indicated concentrations for 24 h. The secreted TNF-α levels were measured by ELISA. (**B**,**C**) B16F10 cells were treated with the CMs of KLK6−/− BMDMs treated with rKLK6 at the indicated conditions for 24 h. The levels of CXCL1 mRNA (**B**) and secreted CXCL1 protein (**C**) were analyzed by RT-PCR and ELISA, respectively. (**D**) KLK6−/− BMDMs were treated with rKLK6 in the presence or absence of the PAR1 antagonist (100 nM). The secreted TNF-α levels were measured by ELISA. (**E**) B16F10 cells were treated with the CMs of indicated KLK6−/− BMDMs for 24 h. The secreted CXCL1 levels were measured by ELISA. ** *p* < 0.01 and * *p* < 0.05 (Student’s *t*-test). Data are representative of three experiments.

**Figure 8 cells-11-04101-f008:**
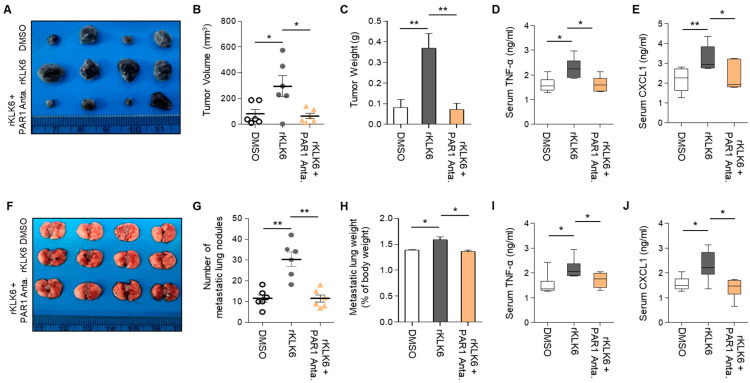
PAR1 inhibitor reduced rKLK6-mediated tumor growth and lung metastasis. (**A**–**E**) B16F10 cells (1 × 10^6^) were subcutaneously injected into KLK6−/− mice. After 1 h, DMSO (control), rKLK6 (10 mg/kg), or rKLK6 plus PAR1 (5 mM/kg) were injected every other day for 10 days. (**A**) Representative photograph of the tumors. The tumor volume (**B**) and tumor weight (**C**) were measured as described in the Materials and Methods. The levels of serum TNF-α (**D**) and CXCL1 (**E**) were measured by ELISA. (**F**,**J**) B16F10 cells (5 × 10^5^) were injected into the tail vein of WT mice. After 1 h, DMSO (control), rKLK6 (10 mg/kg), or rKLK6 plus PAR1 (5 mM/kg) were injected every other day for 10 days. (**F**) Representative photograph of metastatic lung nodules. The number of metastatic lung nodules (**G**) were calculated as described in the Materials and Methods. (**H**) The weight of lungs with metastasized nodules:body weight. The levels of serum TNF-α (**I**) and CXCL1 (**J**) were analyzed by ELISA. ** *p* < 0.01 and * *p* < 0.05 (Student’s *t*-test). Data are representative of three experiments.

## Data Availability

The data presented in this study are available in the Appendix A.

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
