# Peer review of "KLK6/PAR1 Axis Promotes Tumor Growth and Metastasis by Regulating Cross-Talk between Tumor Cells and Macrophages"

_cells, 2022, doi:10.3390/cells11244101_

Round 1

Reviewer 1 Report

* The authors made reasonable efforts to study the KLK6 pathway in the tumor microenvironment but I have some minor issues:

* The abbreviations in the abstract must be defined in their first mention.

* Please, write about CXCL1 and TNF-α in your introduction and their role in the tumor environment with their definition in the first mention. 

* In the material and methods section, please, mention all the catalog numbers for all used kits and instruments.

* Line 96: exchange mouse cancer cell lines with melanoma cell lines.

* Line 169: Please, mention the used antibodies and mention more details about the used technique. 

* I want to suggest the addition of Caspase 3,  Ki67, and VEGF measurement to the western blot of knockout mice to measure the tumor aggressiveness. 

Author Response

The authors made reasonable efforts to study the KLK6 pathway in the tumor microenvironment but I have some minor issues:

Response: We thank reviewer 1 for his/her careful and comprehensive evaluation of our manuscript. We have addressed the concerns suggested by the reviewer and modified the manuscript accordingly. The modified manuscript was highlighted in red. The point-by-point answers to each reviewer’s comments are described below.

1. The abbreviations in the abstract must be defined in their first mention.

Response : We thank the reviewer for making us aware of our error. We have defined abbreviations in the abstract.

2. Please, write about CXCL1 and TNF-α in your introduction and their role in the tumor environment with their definition in the first mention.

Response : As the reviewer suggested, we added additional information about CXCL1 and TNF-α in the introduction section on page 2.

3. In the material and methods section, please, mention all the catalog numbers for all used kits and instruments.

Response : As a reviewer’s comment, we have added all the catalog numbers for all used kits and instruments in the materials and methods section on pages 2 and 3

4. * Line 96: exchange mouse cancer cell lines with melanoma cell lines.

  * Line 169: Please, mention the used antibodies and mention more details about the used technique.

Response : As a reviewer’s comment, we have exchanged melanoma cell lines and added more details about the used antibodies and technique in the material and methods section on pages 2-5.

5. I want to suggest the addition of Caspase 3, Ki67 and VEGF measurement to the western blot of knockout mice to measure the tumor aggressiveness.

Response : We thank the reviewer for the valuable suggestion. We have shown the protein level of caspase 3, Ki67, and VEGF in tumor tissues from WT and KLK6 knockout mice (Figure S2). We have described the result on page 5.

Reviewer 2 Report

In this study the authors show the role of KLK6/PAR1 axis in tumour growth through macrophages-derived TNFα signalling. Using in vivo xenograft models and in vitro tumour-macrophage co-culture model, they show KLK6 knockout mice have slower tumour growth and metastasis. They also show KLK6 promotes TNFα secretion from macrophages via PAR1 in an autocrine manner as well as CXCL1 secretion from tumour cells. They finally prove that inhibition of PAR1 activity could rescue rKLK6 mediated tumour growth in vivo

Overall the authors try to address the question about the role of KLK6/PAR1 axis in tumour growth and metastasis, and could have broader implications in other types of cancers. The study is clearly presented, however, the following issues should be addressed: 

MAJOR CONCERNS

1.  The tumour volume is much less in KLK-/- mice, however, is that because of cell survival issues? Any evidence for that? Since it is tail vein injection, it is not clear whether it is the same cell number to form a tumour. If it is the same cell number to reach the correct position to form a tumour, what is the reason for a smaller tumour? Because of more cell death in KLK6-/- mice? 

2. Since CXCL1 production is critical for tumour growth/metastasis and neutralising antibody could reduce tumour volume, clear evidence is needed to prove the disruption of CXCL1 signalling in the cancer cells. Maybe downstream targets needed to be checked. 

3. PAR1 inhibitor suppressed rKLK6-mediated tumour growth and metastasis. However, you only checked serum CXCL1 level which indicates global change in the mice. It is still necessary to check whether the tumour cell itself also reduces CXCL1 expression. 

4. A further solid experiment is to establish CXCL1 knockdown or knockout stable tumour cell line before injecting into the mice. It will further prove the production of CXCL1 is critical for the KLK6 mediated tumour growth and metastasis. 

5. General histology analysis of the tumour samples is necessary to further evaluate the functional consequence at cellular resolution of KLK6-/- and PAR1 inhibitor. Tumour volume or weight is not sufficient.

Author Response

In this study the authors show the role of KLK6/PAR1 axis in tumour growth through macrophages-derived TNFα signalling. Using in vivo xenograft models and in vitro tumour-macrophage co-culture model, they show KLK6 knockout mice have slower tumour growth and metastasis. They also show KLK6 promotes TNFα secretion from macrophages via PAR1 in an autocrine manner as well as CXCL1 secretion from tumour cells. They finally prove that inhibition of PAR1 activity could rescue rKLK6 mediated tumour growth in vivo. 

Overall the authors try to address the question about the role of KLK6/PAR1 axis in tumour growth and metastasis, and could have broader implications in other types of cancers. The study is clearly presented, however, the following issues should be addressed: 

Response: We thank reviewer 2 for his/her careful and comprehensive evaluation of our manuscript. We have addressed the concerns suggested by the reviewer and modified the manuscript accordingly. The modified manuscript was highlighted in red. The point-by-point answers to each reviewer’s comments are described below.

MAJOR CONCERNS

1. The tumour volume is much less in KLK-/- mice, however, is that because of cell survival issues? Any evidence for that? Since it is tail vein injection, it is not clear whether it is the same cell number to form a tumour. If it is the same cell number to reach the correct position to form a tumour, what is the reason for a smaller tumour? Because of more cell death in KLK6-/- mice?

Response : We thank the reviewer for the supportive comments. We performed western blot analysis to investigate Ki67 and cleaved caspase-3 protein levels in primary tumor tissues from WT and KLK6-/- mice subcutaneously injected with B16F10 cells. Ki67 protein levels were decreased, and cleaved caspase-3 protein levels were increased in tumor tissues from KLK6-/- mice injected with B16F10 cells compared to WT mice (Figure S2). This result suggests that KLK6 may increase the proliferation and reduce apoptosis of cancer cells in vivo. We have described the result on page 5.

2. Since CXCL1 production is critical for tumour growth/metastasis and neutralizing antibody could reduce tumour volume, clear evidence is needed to prove the disruption of CXCL1 signaling in the cancer cells. Maybe downstream targets needed to be checked.

Response : For in vitro experiments, we investigated downstream target protein expression or phosphorylation of signaling mediators to test the activity of the recombinant KLK6, TNF-α neutralizing antibody, CXCL1 neutralizing antibody, and pharmacological inhibitor (PAR1, PAR2). Please find an attached pdf file. For in vivo experiments, we determined the concentration of CXCL1 neutralizing antibody and PAR1 inhibitor by referring to previously reported literature. We have cited these articles in the Materials and Methods section on page 4.

3. PAR1 inhibitor suppressed rKLK6-mediated tumour growth and metastasis. However, you only checked serum CXCL1 level which indicates global change in the mice. It is still necessary to check whether the tumour cell itself also reduces CXCL1 expression.

Response : We are grateful to the reviewer for this comment, and we have investigated the expression level of CXCL1 in tumor tissues from B16F10 melanoma-bearing KLK6 -/-mice treated with DMSO, rKLK6 and rKLK6 plus PAR1 antagonist. rKLK6-injected mice showed enhanced CXCL1 expression compared to DMSO-injected mice (control). However, the enhancement of CXCL1 production was suppressed by the PAR1 inhibitor (Figure S6B). In addition, CXCL1 protein was detected in tumor tissues from WT mice injected with B16F10 melanoma but not in those from KLK6-/- mice injected with B16F10 cells (Figure S6A). Now we have described these results on pages 10 and 13.

4. A further solid experiment is to establish CXCL1 knockdown or knockout stable tumour cell line before injecting into the mice. It will further prove the production of CXCL1 is critical for the KLK6 mediated tumour growth and metastasis. 

Response : The reviewer makes an excellent point. We tried to generate CXCL1 knockdown stable cell lines. However, we could not obtain efficiently CXCL1-depleted stable cell lines. It may be attributable to experimental conditions such as cell types or CXCL1 shRNA. Instead, we used CXCL1-neutralizing antibodies to prove the role of CXCL1 on the KLK6-mediated malignant progression. CXCL1 inhibition by neutralizing antibodies also attenuated rKLK6-promoted tumor growth and lung metastasis (Figure S8). These data support that CXCL1 is critical for KLK6-mediated tumor growth and metastasis. Now we have described these results on page 14.

5. General histology analysis of the tumour samples is necessary to further evaluate the functional consequence at cellular resolution of KLK6-/- and PAR1 inhibitor. Tumour volume or weight is not sufficient.

Response : We thank the reviewer for the supportive comments. We performed the H&E staining in lung tissues in KLK6-/- mice intravenously injected with B16F10 cells and then administered with PBS, rKLK6 alone, or rKLK6 along with the PAR1 inhibitor. We now show the H&E staining data so the reader can see where the cancer cells are (Figure S7). We described this result on page 14.

Reviewer 3 Report

In this manuscript, the author found that KLK6 promoted the production of TNFa from macrophages via PAR1. TNFa secreted by macrophages stimulated CXCL1 production in tumors to promote tumor growth and metastasis. Below are some concerns.

1. Question regarding the mice usage. This is a major concern, the author inoculated tumor cells in the global KLK6 knockout mice. Although the authors showed clear evidence that the tumor size and metastasis changed after KLK6 depletion and also performed a lot of in vitro macrophage-tumor cell co-culture assays, it is still hard to say the phenotype was caused by the macrophage in vivo. It is better to use the macrophage-specific knock-out mice or at least, depletion of macrophage to see if the phenotype is still there.

2. Question regarding Fig.2A-C, Fig. 8E, J. Although CXCL1 was reduced in the serum of the KLK6-KO tumor-bearing mice and also increased when treating mice with rKLK6, no direct evidence shows that the reduction or increase was from the change of CXCL1 in the tumor cells. The author used global KLK6-KO mice, so the alteration of CXCL1 expression may also exist in other cells.

3. Question regarding the CXCL1 expression in tumor cells. The author detected CXCL1 in the serum and tumor cells cultured in the in vitro coculture system. How about the CXCL1 expression in the tumor tissues from tumor-bearing mice?

4. Question regarding the TNFa expression in macrophages. The author detected TNFa expression in the BMDM without any stimulation in vitro. It seems that only the basal level of TNFa was detected by the author. So it can not mimic the tumor microenvironment. The authors need to do FACS to examine the TNFa expression in the macrophages from the tumor tissue.

5. Question regarding CXCL2. The alteration of CXCL2 expression seems not clear. In some charts, CXCL2 showed an obvious reduction on mRNA but not on the protein level, but in the serum, the CXCL2 reduced in the KLK6-KO tumor-bearing mice. The author mainly focused on CXCL1 in this paper, but the proper explanation of CXLC2 should be included in the discussion part.

minor point:

1. Regarding the title of Fig.4. Did the author want to say " KLK6 is responsible for macrophage-promoted CXCL1 production in cancer cells"?

Author Response

In this manuscript, the author found that KLK6 promoted the production of TNFa from macrophages via PAR1. TNFa secreted by macrophages stimulated CXCL1 production in tumors to promote tumor growth and metastasis. Below are some concerns.

Response: We thank reviewer 3 for his/her careful and comprehensive evaluation of our manuscript. We have addressed the concerns suggested by the reviewer and modified the manuscript accordingly. The modified manuscript was highlighted in red. The point-by-point answers to each reviewer’s comments are described below.

1. Question regarding the mice usage. This is a major concern, the author inoculated tumor cells in the global KLK6 knockout mice. Although the authors showed clear evidence that the tumor size and metastasis changed after KLK6 depletion and also performed a lot of in vitro macrophage-tumor cell co-culture assays, it is still hard to say the phenotype was caused by the macrophage in vivo. It is better to use the macrophage-specific knock-out mice or at least, depletion of macrophage to see if the phenotype is still there.

Response : The reviewer makes an excellent point. As the reviewer suggested, we investigated the effect of macrophage depletion on tumor growth. We injected B16F10 cells into WT and KLK6-/- mice and then administered with control or clodronate liposome. Macrophages depletion by clodronate liposome suppressed tumor growth in WT mice. However, this inhibitory effect was reduced in KLK6-/- mice (Figure S4), suggesting that macrophages are, at least partially, important for KLK6-mediated tumor growth. Now we have described these results on page 7.

2. Question regarding Fig.2A-C, Fig. 8E, J. Although CXCL1 was reduced in the serum of the KLK6-KO tumor-bearing mice and also increased when treating mice with rKLK6, no direct evidence shows that the reduction or increase was from the change of CXCL1 in the tumor cells. The author used global KLK6-KO mice, so the alteration of CXCL1 expression may also exist in other cells.

Response : We are grateful to the reviewer for this comment. We investigated the expression of CXCL1 in tumor tissues from B16F10-bearing WT and KLK6-/- mice. CXCL1 protein was detected in tumor tissues from WT mice injected with B16F10 melanoma but not in those from KLK6-/- mice injected with B16F10 cells (Figure S5A). We have also investigated the expression level of CXCL1 in tumor tissues from B16F10 melanoma-bearing KLK6 -/-mice treated with DMSO, rKLK6, and rKLK6 plus PAR1 antagonist. rKLK6-injected mice showed enhanced CXCL1 expression compared to DMSO-injected mice (control). However, the enhancement of CXCL1 production was suppressed by the PAR1 inhibitor (Figure S5B). Moreover, RT-PCR showed that CXCL1 mRNA levels in B16F10 melanoma significantly increased on coculture with WT BMDMs but not with KLK6-/- BMDMs (Figure 3E). CXCL1 mRNA expression was not detected in WT and KLK6-/- BMDMs on coculture with cancer cells. (Figure S5). We have now described these results on pages 8, 10, and 13.

3. Question regarding the CXCL1 expression in tumor cells. The author detected CXCL1 in the serum and tumor cells cultured in the in vitro coculture system. How about the CXCL1 expression in the tumor tissues from tumor-bearing mice?

Response: We thank the reviewer for the supportive comments. We investigated the expression of CXCL1 protein in tumor tissues from tumor-bearing mice. Western blot analysis showed that CXCL1 protein was detected in tumor tissues from WT mice injected with B16F10 melanoma but not in those from KLK6-/- mice injected with B16F10 cells (Figure S5A). We have also investigated the expression level of CXCL1 in tumor tissues from B16F10 melanoma-bearing KLK6 -/-mice treated with DMSO, rKLK6, and rKLK6 plus PAR1 antagonist. rKLK6-injected mice showed enhanced CXCL1 expression compared to DMSO-injected mice (control). However, this enhancement was suppressed by the PAR1 inhibitor (Figure S5B). We have now described these results on pages 10 and 13.

4. Question regarding the TNFa expression in macrophages. The author detected TNFa expression in the BMDM without any stimulation in vitro. It seems that only the basal level of TNFa was detected by the author. So it can not mimic the tumor microenvironment. The authors need to do FACS to examine the TNFa expression in the macrophages from the tumor tissue.

Response : As reviewer’s comment, macrophage cells in primary tumor sites were sorted using CD11b+/F4/80+ macrophage maker and subjected to flow cytometry and western blot analysis to investigate the TNF-α expression. The levels of TNF-α were increased in macrophages isolated from tumor tissues of WT mice injected with B16F10 cells compared to KLK6-/- mice (Figure S9). We have described these results on page 14.

5. Question regarding CXCL2. The alteration of CXCL2 expression seems not clear. In some charts, CXCL2 showed an obvious reduction on mRNA but not on the protein level, but in the serum, the CXCL2 reduced in the KLK6-KO tumor-bearing mice. The author mainly focused on CXCL1 in this paper, but the proper explanation of CXCL2 should be included in the discussion part.

Response : We thank the reviewer for the supportive comments. CXCL1 mRNA levels and secretion in B16F10 melanoma significantly increased on coculture with WT BMDMs but not with KLK6-/- BMDMs (Figure 3E and 3F). Moreover, CXCL1 mRNA expression was not detected in both WT and KLK6-/- BMDMs on coculture with cancer cells. However, CXCL2 mRNA expression increased in WT/KLK6-/- BMDMs and cancer cells when cocultured (Figure S5). These results suggest that KLK6 stimulates CXCL1 expression in cancer cells when cocultured with BMDMs, whereas it does not affect CXCL2 expression in cancer cells and BMDMs. Therefore, we mainly focused on CXCL1 for further experiments. Now we described these results on pages 8 and 16.

minor point: point:                                                                                                   

1. Regarding the title of Fig.4. Did the author want to say "KLK6 is responsible for macrophage-promoted CXCL1production in cancer cells"?

Response: We thank the reviewer for making us aware our error. Now we have corrected the error.

Round 2

Reviewer 2 Report

The authors have addressed my concerns. It can be accepted in the present format. 

Reviewer 3 Report

The authors well addressed the concerns and the current form is OK to publish.